# Enhancing *Dengue* Virus *Production and Immunogenicity* with Celcradle™ Bioreactor: A Comparative Study with Traditional Cell Culture Methods

**DOI:** 10.3390/vaccines12060563

**Published:** 2024-05-22

**Authors:** Hongxia Guo, Xiaoyan Ding, Dong Hua, Minchi Liu, Maocheng Yang, Yuanxin Gong, Nan Ye, Xiaozhong Chen, Jiuxiang He, Yu Zhang, Xiaofeng Xu, Jintao Li

**Affiliations:** 1Department of Biosafety, School of Basic Medicine, Army Medical University, Chongqing 400038, China; hongxiaguo@tmmu.edu.cn (H.G.); dxywork2012@sina.cn (X.D.); huadong@tmmu.edu.cn (D.H.); minchiliu@tmmu.edu.cn (M.L.); mchy@tmmu.edu.cn (M.Y.); gongyx0932@tmmu.edu.cn (Y.G.); nan.ye@gcp-clinplus.com (N.Y.); chenxiaozhong@tmmu.edu.cn (X.C.); xiang18@tmmu.edu.cn (J.H.); zhangyu_tmmu@tmmu.edu.cn (Y.Z.); xfxu@tmmu.edu.cn (X.X.); 2Department of Pediatrics, Ludwig-Maximilians University of Munich, 80337 Munich, Germany

**Keywords:** dengue virus, bioreactor, vaccine, immunogenicity

## Abstract

The dengue virus, the primary cause of dengue fever, dengue hemorrhagic fever, and dengue shock syndrome, is the most widespread mosquito-borne virus worldwide. In recent decades, the prevalence of dengue fever has increased markedly, presenting substantial public health challenges. Consequently, the development of an efficacious vaccine against dengue remains a critical goal for mitigating its spread. Our research utilized Celcradle™, an innovative tidal bioreactor optimized for high-density cell cultures, to grow Vero cells for dengue virus production. By maintaining optimal pH levels (7.0 to 7.4) and glucose concentrations (1.5 g/L to 3.5 g/L) during the proliferation of cells and viruses, we achieved a peak Vero cell count of approximately 2.46 × 10^9^, nearly ten times the initial count. The use of Celcradle™ substantially decreased the time required for cell yield and virus production compared to conventional Petri dish methods. Moreover, our evaluation of the immunogenicity of the Celcradle™-produced inactivated DENV4 through immunization of mice revealed that sera from these mice demonstrated cross-reactivity with DENV4 cultured in Petri dishes and showed elevated antibody titers compared to those from mice immunized with virus from Petri dishes. These results indicate that the dengue virus cultivated using the Celcradle™ system exhibited enhanced immunogenicity relative to that produced in traditional methods. In conclusion, our study highlights the potential of the Celcradle™ bioreactor for large-scale production of inactivated dengue virus vaccines, offering significant promise for reducing the global impact of dengue virus infections and accelerating the development of effective vaccination strategies.

## 1. Introduction

The dengue virus (DENV) represents the most widespread mosquito-borne virus globally, transmitted predominantly through the bites of Aedes albopictus and Aedes aegypti mosquitoes. It is composed of four distinct serotypes (DENV1, DENV2, DENV3, and DENV4), each capable of inducing a range of illnesses from dengue fever (DF) to more severe forms such as dengue hemorrhagic fever (DHF) and dengue shock syndrome (DSS) [1,2]. Annually, an estimated 390 million infections worldwide highlight the substantial public health challenge posed by DENV [1]. Data from the World Health Organization (WHO) indicate a marked escalation in dengue cases over recent decades, affecting nearly half the global population [3]. The absence of specific therapeutic or preventive options for dengue underscores the critical need for the development of effective vaccines to mitigate the spread of this virus.

Up to date, two vaccines against the dengue virus have been licensed: CYD-TDV (Dengvaxia) and TAK-003 (Qdenga). CYD-TDV is specifically recommended for individuals with prior dengue infection, as it can increase the risk of severe dengue in those not previously infected through a mechanism known as antibody-dependent enhancement (ADE) [4]. TAK-003 shows potential in reducing hospitalizations and symptoms associated with dengue, yet its efficacy varies among the different dengue serotypes, and its long-term safety and efficacy in individuals without prior exposure to the virus are still being evaluated [5]. Both vaccines have restricted indications and are advised mainly for those previously infected and residing in endemic regions, limiting their applicability to the general populace, including travelers [4,6]. Challenges persist with live-attenuated tetravalent dengue vaccines, including inconsistent viral replication and biased neutralizing antibody responses influenced by pre-existing immunity to DENV and other flaviviruses [7,8,9]. Consequently, research is directed towards alternative approaches, such as nonreplicating inactivated virus vaccines, which promise greater safety, stability, and ease of maintenance [7,10]. Previous research has underscored the efficacy of a formalin-treated quadrivalent inactivated dengue vaccine produced using Vero cells, which elicited robust and sustained neutralizing antibody titers in rhesus monkeys [11]. Ongoing advancements in process technology, virus propagation methodologies, cell lines, and adjuvant formulations continue to enhance the prospects for developing efficacious inactivated dengue vaccines.

Bioreactors are integral to the production of viral vaccines on a large scale using cell culture technologies, and they are crucial in the manufacturing processes for a variety of viral vaccines, such as those for SARS-CoV-2, Zika, influenza, Enterovirus, and Rabies virus [12]. Among the innovative developments in this field is the Celcradle™ bioreactor system, which features a fixed bed and a specially designed Celcradle^TM^ bottle for high-density cell culture. The system’s design includes an upper chamber equipped with a carrier bed and a lower chamber filled with medium contained within compressive bellows. The dynamic oscillation of the bellows facilitates a cyclic process where immobilized cells are alternately submerged in the medium to access nutrients and then exposed to ambient air to allow gas exchange. This scalable packed-bed bioreactor system is not only straightforward to operate but also provides a cost-effective method for large-scale vaccine production. Recently, Offersgaard A et al. utilized this bioreactor to cultivate Vero cells and produce the SARS-CoV-2 virus without the use of animal components [13]. Vero cells, commonly employed in vaccine production, are particularly significant as most dengue vaccines undergoing clinical trials utilize these cells [14,15].

At the outset of this study, there were no documented studies detailing the utilization of bioreactors for the production of dengue virus vaccines. Consequently, the objective of this study was to develop a methodology for the production of a whole virus dengue vaccine using Vero cells within the Celcradle™ bioreactor system and to evaluate the immunogenicity of the viral antigen. The ability to produce viruses efficiently and at scale in cell lines is imperative not only for the manufacturing of inactivated and attenuated whole virus vaccines but also plays a vital role in supporting associated animal immunogenicity studies, vaccine efficacy challenges, and diagnostic assays.

## 2. Materials and Methods

### 2.1. Cell Line, Virus Strain, and Bioreactor

Vero cells (CRL-1586) were procured from the American Type Culture Collection (ATCC) and stored in our laboratory. The dengue virus strain, specifically adapted for Vero cells, was provided by the Guangzhou Center for Disease Control and Prevention, Guangzhou Military Command (Guangzhou, China). All procedures involving live viruses were performed within a biosafety cabinet situated in a Biosafety Level 2 (BSL-2) laboratory to ensure stringent adherence to safety protocols. The bioreactor system employed for the viral cultivation was the Celcradle™, a tidal principle fixed-bed bioreactor provided by Esco Lifesciences Group, based in Singapore.

### 2.2. Cell Culture in Petri Dishes

Adherent Vero cells were initially cultured in Petri dishes (Corning, 430599) using 20 mL of DMEM high glucose medium (Gibco, Life Technologies, Carlsbad, CA, USA, #C11995500BT) supplemented with 5% Fetal Bovine Serum (FBS) (Gibco, Life technologies, USA, #10270-106), and 1% Penicillin and Streptomycin (Beyotime, Shanghai, China, C0222). The process of cell passage cultivation was performed as follows: after two washes with phosphate-buffered saline (PBS) (Biosharp, Guangzhou, China, #BL302A), a thin layer of 0.25% trypsin solution (Biosharp, #BL501A) was added to the monolayer cells, and the cells were incubated for 3 min in a 37 °C cell culture tank (Esco Lifesciences Group, Singapore). Subsequently, the detached cells were centrifuged at 180 g for 5 min, resuspended in fresh culture medium, and transferred to new Petri dishes for further cultivation. Static cultures were maintained in a cell incubator set to 37 °C with 5% CO_2_.

### 2.3. Infection and Proliferation of DENV4 in Petri Dishes

DENV4 was diluted in serum-free DMEM medium and introduced to a monolayer of Vero cells after washing the cells twice with PBS (8 mL/dish). The cells were incubated with the DENV4 inoculum for 2 h at a multiplicity of infection (MOI) value of 0.001. After removing the DENV4 inoculum, 20 mL of DMEM supplemented with 2% FBS and 1% penicillin, and streptomycin was added. The cell was maintained for 7 days, and the virus supernatant was collected.

### 2.4. Adaptation of Vero Cells to the Celcradle™ Bioreactor System

As shown in Figure 1, approximately 3.0 × 10^8^ Vero cells detached from cell culture dishes were resuspended in 100 mL of DMEM medium. This cell suspension was then introduced into the Celcradle^TM^ bioreactor bottle, and the white airtight lid was secured. The Celcradle^TM^ bottle was placed upside down in a 37 °C incubator to ensure the carriers that BioNOC^TM^ II microcarriers, where cells attach, were fully immersed in the cell suspension for 1.5 h. Gentle agitation was applied every 30 min. Subsequently, 400 mL of DMEM containing 5% FBS and 1% penicillin, and streptomycin was added to the Celcradle^TM^ bottle, and the white lid was exchanged for a blue breathable lid equipped with a 0.22μm filter membrane. The bioreactor bottle was then mounted on the stage, and operating parameters were set as follows: an up rate of 1.5 mm/s and T_H: 10 s and a down rate of 1.5 mm/s and B_H: 30 s. Cell division was monitored daily, and the pH and glucose content of the medium were regularly assessed.

### 2.5. DENV4 Infection and Proliferation in the Celcradle™ Bioreactor System

The Vero cells were initially washed with 100 mL of serum-free DMEM medium, followed by the inoculation of 500 mL of DENV4 inoculum at an MOI of 0.001. The virus infection protocol was set with an up rate of 2.0 mm/s and T_H: 20 s and a down rate of 2.0 mm/s and B_H: 0 s. Subsequently, 10 mL of FBS was added to the bottle, and the operational procedure was adjusted to an up rate of 1.5 mm/s, T_H: 10 s and a down rate of 1.5 mm/s and B_H: 30 s. Daily samples were taken for nucleic acid detection, and continuous monitoring of cell division, pH, and glucose levels in the medium was conducted.

### 2.6. Monitoring of Glucose

The glucose levels in the culture medium were assessed using the GlucCell^TM^ detector. When glucose levels approached 1 g/L, the spent medium was replaced with a fresh DMEM medium containing 5% FBS and 1% penicillin and streptomycin.

### 2.7. Monitoring of pH

The pH values in the culture medium were performed using a pH meter. If the pH values fell below or approached 7.0, 7.5% NaHCO_3_ was added to the medium to adjust the pH to approximately 7.4.

### 2.8. Monitoring of Cell Count

For cell counting, two carriers were extracted from the bottle and placed into a 1.5 mL centrifuge tube containing 1 mL of crystal violet dye (CVD) nuclear counting kit (Cat No.:1400014). The tubes were vibrated for 1 min and then incubated at 37 °C for 1 h with intermittent vibrations every 15 min for 5 s to release the nuclei. Next, a 20 μL sample of the suspension was then used for cell counting on a Countstar biotech plate.

### 2.9. Monitoring of Virus RNA

Daily, virus suspension and carriers were collected in 1.5 mL tubes. The materials underwent three freeze–thaw cycles at −80 °C and room temperature to liberate the virus. Post freeze–thaw, the suspension was centrifuged at 10,000× *g* for 30 min at 4 °C to remove cell debris, yielding pure virus supernatant. RNA extraction utilized 200 μL of the supernatant with a Takara RNA extraction kit (9766), followed by reverse transcription using a Takara kit (RR047A). RT-PCR utilized primers with sequences 5’-atggtggaagccttgtcagatgc-3’ and 5’-tcggtttttctgtttcagtgga-3’, conducted over 45 cycles on a Roche LightCycler96. A positive plasmid of DENV4 was employed to establish a standard curve, and RNA copy numbers were quantified using the method described in the reference [16].

### 2.10. Purification and Concentration of Virus and Detection of Protein

The virus fluid underwent centrifuging at 10,000× *g* for 30 min to eliminate cell debris, followed by the addition of an equal volume of 16% PEG6000and overnight mixing at 4 °C. The virus precipitate was then obtained by further centrifugation at 10,000× *g* for 2 h (Beckman, Guangzhou, China). This precipitate was resuspended in PBS and aliquated for storage at −80 °C. For protein concentration detection, 20 μL of the virus was mixed with 80 μL of RIPA Lysis Buffer (Beyotime, P0013B), vortexed 1 min, lysed on ice for 15 min, and centrifuged at 4 °C for 15 min. Protein concentrations were subsequently measured using the Enhanced BCA Protein Assay Kit (Beyotime, P0010S) after a 30 min incubation in a 56 °C water bath.

### 2.11. Animal Immunization

The inactivated DENV4 vaccine was obtained by heating the virus at 56 °C for 30 min by mixing with TH-Z93 adjuvant [17], a lipophilic bisphosphonate developed by Professor Zhang Yonghui’s team of Tsinghua University, and suspension in PBS to a final injection volume of 200 μL. Female BALB/c mice, 6 weeks old, were obtained from the Animal Center of Army Medical University and housed under controlled conditions (24 °C temperature and 12 h light–dark cycle) with free access to water and standard laboratory food. Mice were acclimatized to the environment for 7 days before immunization. The mice were divided into four groups (*n* = 5 per group): Two groups were immunized with the bioreactor-derived and Petri dish-derived DENV4 inactivated vaccines, respectively, receiving three doses biweekly via subcutaneous injection in the neck using 100 μg of vaccine. Two control groups received equivalent volumes of adjuvant or PBS. Blood samples (30 μL) were collected from the tail vein preimmunization and on day 14 post of the final immunization. The serum was separated by centrifuging the blood at 2900× *g* for 10 min at 4 °C (Eppendorf, Hamburg, Germany).

### 2.12. Detection of Antibodies Titer

Serum titers were evaluated using an indirect enzyme-linked immunosorbent assay (ELISA). Each well of a 96-well flat plate (CORNING, Corning, NY, USA, 9018) was coated with 100 μL of a solution containing 1.25 μg of inactivated DENV4 and incubated overnight at 4 °C. Subsequently, 200 μL/well of serum diluted in 2% FBS/PBS was added and incubated at 37 °C for two hours. After three washes with 200 μL/well of 0.05% Tween20/PBS, 100 μL/well of HRP-conjugated IgG antibody (1:5000 dilution) (Solarbio, Beijing, China, 20210329) was added, and the plate incubated for another 40 min at 37 °C. Following the removal of unbound antibodies, 100 μL/well of TMB solution was added and incubated for 15 min at 37 °C. The reaction was stopped by adding 50 μL of 2 M sulfuric acid to each well, and absorbance was measured at 450 nm (Tecan).

### 2.13. Statistical Analysis

Data were analyzed using GraphPad Prism 9.5 (GraphPad Software Inc., San Diego, CA, USA, 2022). Statistical significance was defined at *p*-values of <0.05 (*), <0.01 (**), <0.001 (***), and <0.0001 (****). One-way ANOVA was employed for multiple comparisons to assess the differences in RNA copies post-infection between groups receiving primary doses of supernatant and cells on carriers. Two-way ANOVA was utilized to evaluate differences in immunogenicity and efficacy between the experimental groups.

## 3. Results

### 3.1. Adaptive Growth of Vero Cells on Suspended Carriers in the CelCradle^TM^ Bioreactor System

Vero cells, known for their utility in vaccine production due to susceptibility to various viruses [18,19,20], are particularly integral to the production of dengue virus vaccines [21,22]. In this study, the adaptation of Vero cells to suspended growth within the CelCradle™ bioreactor system was closely monitored to develop an optimal cell culture protocol that supports continuous viral multiplication. Initially, Vero cells were transitioned from Petri dish cultures to suspended growth in the CelCradle™ system, as depicted in Figure 1, with a starting culture volume of 500 mL. The optimal initial density was determined to be approximately 3 × 10^8^ cells in 100 mL of medium (equating to 3 × 10^6^ cells/mL). Following 1.5 h of incubation in an inverted position within the cell incubator, an 85% cell adsorption rate was achieved. An additional 400 mL of 5% FBS/DMEM was then introduced into the bioreactor, and the culture’s media and gas exchanges were managed according to the protocol illustrated in Figure 2A. Glucose levels were maintained at optimal concentrations, and medium exchanges began on the second day of culture, as shown in Figure 2B. To ensure cell viability, the pH of the medium was consistently monitored and adjusted to maintain a range of 7.0 to 7.4, using carbon dioxide adjustments (Figure 2C) and periodic additions of NaHCO_3_ (Figure 2D). By the seventh day, cell numbers peaked at approximately 2.49 × 10^9^, as recorded in Figure 2E, underscoring the successful adaptation of Vero cells to high-density suspension growth in the CelCradle™ bioreactor system. This adaptation facilitates effective cell proliferation, which is essential for scalable viral vaccine production.

The entire CelCradle™ bioreactor system is enclosed within a 37 °C carbon dioxide cell incubator. The process begins with the resuscitation and expansion of Vero cells in Petri dishes (A). Subsequently, either a Vero cell suspension or a virus solution is transferred into the CelCradle™ bottle. The fluid level is adjusted to ensure all carriers are submerged, facilitating cell adherence to fibers or virus infection of cells (B). During the operation, the descending movement of the holding plate causes the medium to drop onto the lower bellows, exposing the carriers to air and facilitating oxygen transfer (C). Following a brief delay, the ascending movement of the holding plate compresses the bellows, raising the medium level to resubmerge the carriers, thus promoting nutrient transfer (D). The cyclic change in the rotating direction of the screw shaft drives the holding plate in an up-and-down motion, effectively modulating the environment for optimal cell growth and viral infection.

### 3.2. DENV Proliferation through Vero Cells in the CelCradle^TM^ Bioreactor System

Following the successful adaptation of Vero cells to suspension growth in the CelCradle™ bioreactor system, efforts shifted towards the mass production of DENV. The DENV infection of the adapted cells was initiated on day 7 of the suspension culture (Figure 3A). Glucose content and pH levels of the medium were meticulously monitored and adjusted daily, maintaining glucose between 1.5 g/L and 3.5 g/L (Figure 3B) and pH between 7.0 and 7.4 (Figure 3C). The cell count on the fiber carriers was tracked continuously. Initially, during the adaptive growth phase, the cell numbers increased; however, post-infection, there was a gradual decrease in cell count (Figure 4A). HE staining revealed that Vero cells on the fiber carriers began to detach following viral infection (Figure 4B). To assess virus production further, DENV nucleic acid levels were analyzed. Intracellular viral nucleic acids showed a significant increase, peaking on day 9 (Figure 5A), while nucleic acids in the supernatant similarly peaked on day 9 (Figure 5B). These findings demonstrate the effective production of DENV at high titers by Vero cells in the CelCradle™ bioreactor system, establishing a viable bioreactor process for DENV vaccine production.

### 3.3. Immunogenicity of DENV Produced by the CelCradle^TM^ Bioreactor System

To evaluate the immunogenicity of the produced DENV, BALB/c mice were subcutaneously immunized with 100 μg of inactivated DENV derived from both the CelCradle™ bioreactor system and traditional Petri dishes (via a 56 °C water bath) (Figure 6A). Serum titers indicated that the mice immunized with DENV produced in the CelCradle™ bioreactor system exhibited higher immunogenicity compared to those immunized with the DENV from Petri dishes. Moreover, the antiserum from mice demonstrated cross-reactivity with inactivated viruses derived from both the bioreactor and the Petri dishes, as shown in Figure 6B–G. These results suggest that the bioreactor-derived DENV4 not only triggers a robust immune response in BALB/c mice but also shows a superior immune effect compared to the conventionally derived DENV4 from Petri dishes.

## 4. Discussion

Efficient virus production is crucial for whole virus vaccines [13]. In this study, we have demonstrated the feasibility and efficiency of using suspension-adapted Vero cell cultures in the Celcradle^TM^ bioreactor system for the production of the DENV. Throughout the experimentation, continuous monitoring of glucose, pH, and cell numbers in the medium facilitated optimal cell growth and virus proliferation. Additionally, we have validated the immunogenicity of the inactivated DENV produced by the CelCradle^TM^ bioreactor system in mice, where those immunized with the bioreactor-derived DENV exhibited higher serum antibody titers compared to those receiving the Petri dish-derived DENV. This represents the first successful report of DENV production through a bioreactor system.

Vero cells, being the first approved continuous cell line for human viral vaccine production, have historically been employed in the manufacture of vaccines against influenza, rabies, poliovirus, and rotavirus [23,24,25]. Their adaptability to suspension culture makes them particularly suitable for the production of live attenuated or inactivated vaccines for dengue viruses [26,27,28]. Vero cells can be cultured in suspension and used to manufacture other viral vaccines [14,29]. Therefore, in this study, we first cultured Vero cells in suspension in the CelCradle^TM^ bioreactor system, achieving a 10-fold increase in cell growth compared to the initial density. Unlike conventional cell factories or roller bottles, which are prone to contamination risks, poor oxygen delivery, and limited surface area, the CelCradle™ system, equipped with BioNOC™ microcarriers, supports efficient, adherence-dependent cell growth in both serum-containing and serum-free media. These microcarriers offer superior liquid mixing and fixation efficiency, shear force protection, and nutrient transfer, enhancing cell culture outcomes [13,30,31].

The Celcradle^TM^ bioreactor system has been instrumental in supporting the production of various viruses, including Japanese encephalitis virus, hepatitis D virus-like particles, influenza A virus, small ruminant morbillivirus, rift valley fever virus, lumpy skin disease virus, and the current pandemic SARS-CoV-2 [11,13,32,33,34,35]. In this study, we successfully produced DENV using suspension-adapted Vero cells in the CelCradle^TM^ bioreactor system. The successful replication of viral nucleic acid in the cells and the significant increase in viral nucleic acid levels in the culture supernatant indirectly reflected the multiplication of the DENV in suspension-cultured Vero cells using the CelCradle^TM^ bioreactor system. During cell suspension culture and viral proliferation, we monitored glucose and pH levels in the medium daily, as these can significantly impact cell growth, productivity, and virus characteristics, such as virus membrane fusion, virus maturation, stability, and infectivity [36,37]. During the process, we maintained the glucose concentration between 1.5 g/L and 3.5 g/L and the pH between 7.0 and 7.4 through regular medium replacements or by adding glucose and NaHCO_3_ solution. Although the CelCradle^TM^ bioreactor system offers convenient pH regulation and nutrient addition, maintaining pH and energy supply within the desired range manually can be challenging. However, improved versions of the Celcradle^TM^ and the larger-scale TideXcell systems are now equipped with closed-loop pH control, which enables stable pH maintenance during cell culture and virus propagation, further enhancing productivity [13]. Cell culture strategies in the bioreactor can be further optimized through the regulation of medium formulation, additive addition, partial medium exchange, and medium recirculation or perfusion during cultivation to maintain nutrient levels and reduce waste accumulation, ultimately improving cell and virus productivity [31,38,39,40]. The collection of viruses also needs to be further explored, and strategies need to be refined.

Furthermore, the DENV produced in the Celcradle^TM^ bioreactor system was purified, inactivated, and injected into mice to assess its immunogenicity as a vaccine. Mice immunized with the DENV produced using the Celcradle^TM^ bioreactor system showed higher antibody titers than those immunized with the Petri dish-derived DENV. Moreover, the antiserum from mice immunized with the DENV cross-reacted with antigens from both the bioreactor and Petri dish-derived DENV, and the antiserum could cross-react with antigens from the two sources, indicating similar characteristics and immunogenicity. These results confirm that the Celcradle^TM^ bioreactor system culture is a promising new method for dengue virus proliferation and can be utilized for the manufacture of dengue vaccines.

## 5. Conclusions

In this study, we successfully established an efficient production method for the DENV using the innovative Celcradle^TM^ bioreactor system, a scalable and adaptable cell culture technology particularly suited for process development and optimization in vaccine production. We demonstrated that the immunogenicity of the DENV derived from the Celcradle^TM^ bioreactor system proved to be superior to that of the virus derived from the traditional Petri dish method. The findings have significant implications for the development of whole virus vaccines against dengue, highlighting the Celcradle™ bioreactor system as a promising platform for large-scale virus production. This advancement could greatly enhance research on viral pathogenesis and accelerate the development of effective vaccines, potentially impacting global health by improving response capabilities against dengue outbreaks.

## Figures and Tables

**Figure 1 vaccines-12-00563-f001:**
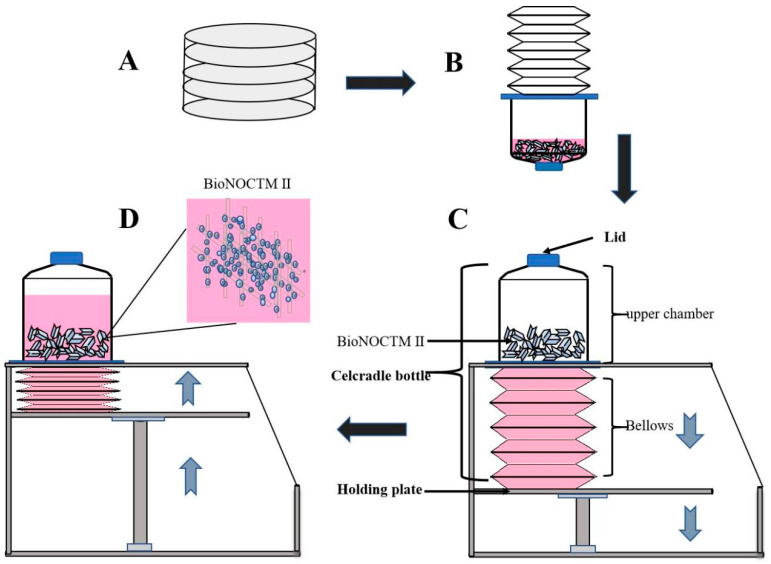
Schematic representation of the Celcradle^TM^ bioreactor system. The entire CelCradle™ bioreactor system was enclosed within a 37 °C carbon dioxide cell incubator. The process began with the resuscitation and expansion of Vero cells in petri dishes (**A**). Subsequently, either a Vero cell suspension or a virus solution was transferred into the CelCradle™ bottle. The fluid level was adjusted to ensure all carriers were submerged, facilitating cell adherence to fibers or virus infection of cells (**B**). During the operation, the descending movement of the holding plate caused the medium to drop onto the lower bellows, exposing the carriers to air and facilitating oxygen transfer (**C**). Following a brief delay, the ascending movement of the holding plate compressed the bellows, raising the medium level to re-submerge the carriers, thus promoting nutrient transfer (**D**). The cyclic change in the rotating direction of the screw shaft drived the holding plate in an up and down motion, effectively modulating the environment for optimal cell growth and viral infection.

**Figure 2 vaccines-12-00563-f002:**
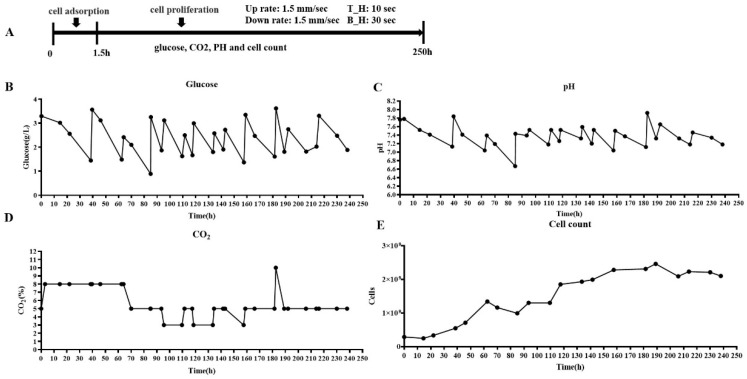
**Adaptive growth of adherent Vero cells on suspended carriers in the Celcradle^TM^ bioreactor system.** The programmed cycle for Vero cell proliferation (**A**), alongside the continuous monitoring of key environmental and nutritional parameters such as glucose levels (**B**), pH of the medium (**C**), CO_2_ concentration in the incubator (**D**), and the resultant cell count (**E**).

**Figure 3 vaccines-12-00563-f003:**
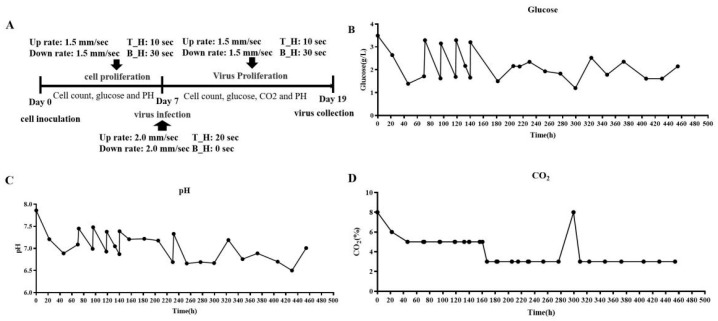
The DENV proliferation and monitoring of pH, glucose, and CO_2_ during the process in the Celcradle^TM^ bioreactor system. DENV4 multiplication program (**A**) tracks the monitoring of critical growth factors, including glucose (**B**), pH levels in the medium (**C**), and CO_2_ concentration in the incubator (**D**) throughout the viral proliferation process.

**Figure 4 vaccines-12-00563-f004:**
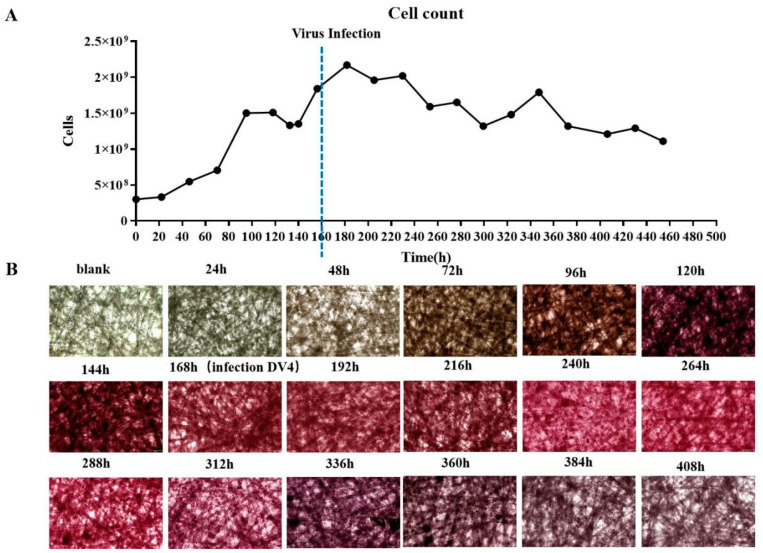
The monitoring of Vero cell counts during DENV proliferation in the Celcradle^TM^ bioreactor system. The dynamic changes in Vero cell counts during the DENV proliferation process (**A**) provide histological analysis through HE staining of Vero cells on the carriers (**B**), illustrating cellular morphology and density.

**Figure 5 vaccines-12-00563-f005:**
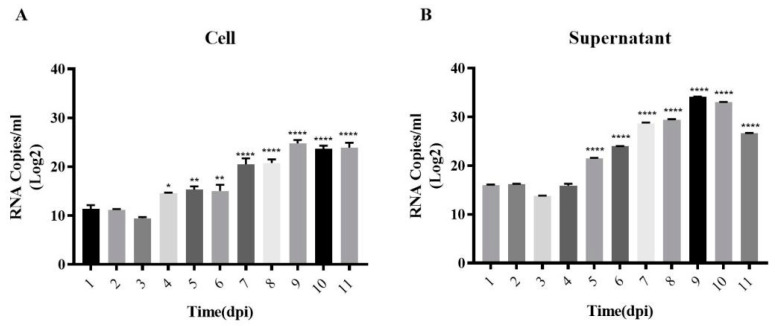
**Nucleic acid monitoring during DENV proliferation in the Celcradle^TM^ bioreactor system.** The changes in viral nucleic acid, including the intracellular DENV nucleic acid on carriers (**A**) and virus in medium (**B**). The *p*-values label (* indicates *p* < 0.05, ** indicates *p* < 0.01, and **** indicates *p* < 0.0001) represents the significance of the RNA copies from day to day compared to the first day of the process.

**Figure 6 vaccines-12-00563-f006:**
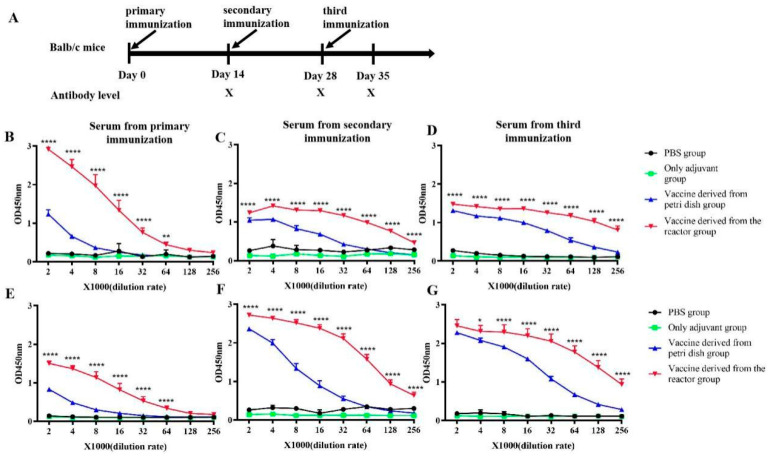
**The titer and cross reactivity of antisera of the mice immunized with inactivated DENV vaccine.** Immunological processes in BALB/c mice (**A**). The titer of immune sera against DENV from the Celcradle^TM^ bioreactor system (**B**–**D**). The titer of immune sera against DENV from the Petri dish (**E**–**G**). The *p*-value labels (* indicates *p* < 0.05, ** indicates *p* < 0.01, and **** indicates *p* < 0.0001) statistically significant differences between the vaccine batches derived from the reactor and those from the Petri dish, providing insights into the efficacy and potency of the vaccine produced by each method.

## Data Availability

The data presented in this study are available on request from the corresponding author.

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
