# Peer review of "Enhancing Dengue Virus Production and Immunogenicity with Celcradle™ Bioreactor: A Comparative Study with Traditional Cell Culture Methods"

_vaccines, 2024, doi:10.3390/vaccines12060563_

Round 1
Reviewer 1 Report
Comments and Suggestions for Authors
Reviewer’s Summary of the Manuscript, entitled “Celcradle™ Bioreactor Enhances Dengue Vaccine Production and Immunogenicity” that the use of a CelcradleTM”
Reviewer's comment highlighted in yellow and under-lined
The authors describe their study entitled, “Celcradle™ Bioreactor Enhances Dengue Vaccine Production and Immunogenicity” that the use of a CelcradleTM or a novel bioreactor that was designed to increase the concentration of Vero cells to support the replication of an increased concentration of dengue 4 (DEN 4) virus for producing an inactivated DENV vaccine. According to the authors, the immunogenicity of the Celcradle™-produced an inactivated DV4 vaccine that was assessed by immunizing mice. The vaccine elicited higher antibody titers than in mice immunized with petri dish-cultured vaccine antigens. The findings suggested that the dengue virus produced using the Celcradle™ system possesses equivalent or superior immunogenicity to that from petri dishes. In summary, our findings shed light on Celcradle™ bioreactor's potential for large-scale production of inactivated dengue virus vaccines. This advancement holds promise for mitigating the global burden of dengue virus infection and expediting the development of effective vaccination strategies.
Comments on the Abstract
Abbreviate as suggested by the reviewer throughout the manuscript as in this first sentence of the Abstract: The Dengue virus, (should be dengue viruses and DENV should replace the use of DV) responsible for dengue fever (DF) , dengue hemorrhagic fever (DHF) , and dengue shock syndrome (DSS) , ranks as the most prevalent mosquito-borne virus globally. The incidence of DF has surged significantly in recent decades, posing major public health challenges.
In the abstract, the reader is led to believe that the use of the Celcradle increased the concentration of cells and the yield of virus, but the following sentence implies otherwise “Utilizing Celcradle™ drastically reduced cell yield and virus production times compared to traditional petri dish methods”
These findings suggest (should be suggested) that the dengue virus produced using the Celcradle™ system possesses (should be possessed) equivalent or superior immunogenicity to that from petri dishes. In summary, our findings shed light on Celcradle™ bioreactor's potential for large-scale production of inactivated dengue virus vaccines.
In summary, our findings shed (should be shedded) light on Celcradle™ bioreactor's potential for large-scale production of inactivated dengue virus vaccines. This advancement holds promise for mitigating the global burden of dengue virus infection and expediting the development of effective vaccination strategies.
Comments on the Introduction
The World Health Organization (WHO) statistics reveal (should be revealed) a substantial increase in dengue fever cases in recent decades, posing a threat to approximately half of the world's population[3].
Comments on Materials and Methods
Section 2.2 The dislodged cells were then centrifuged at 1000 rpm (convert to g) for 5 minutes and re-suspended in culture medium before being placed in new
Section 2.3 DV4 was diluted with serum-free DMEM medium and added to monolayer of Vero cells after washing the cells twice with PBS (8 ml/dish). The cells were incubated with the DV4 solution (use inoculum rather than solution) for 2 hours at an MOI (Multiplicity of Infection) (should Multiplicity of Infection and then abbreviate as MOI) value of 0.001. After removing the DV4 (should be inoculum rather than solutions) solution, 20mL of DMEM supplemented with 2% FBS and 1% Penicillin and Streptomycin was added. The cells were cultured (cell were maintained for 7 days, and the virus supernatant was collected (add collected from the infected cells).
Section 2.4 What was the concentration of cells added to the Celcradle bottles and an s should be added to bottle. “As shown in Fig. 1, Vero cells were detached from cell culture dishes and suspended in 100 mL of DMEM medium. The cell suspension was added to the Celcradle bottle, and the white airtight lid was closed”.
What is meant by the use of the word “carrier” in the following sentence.
“The Celcradle bottle was then placed upside down in a 37℃ cell incubator, ensuring that the carrier was fully immersed in the cell suspension for 3 hours, with gentle shaking every 30 minutes”.
Section 2.5 The cells were washed with 100 mL of serum-free DMEM medium. Then, 500 mL of DV4 solution (should be inoculum) was inoculated with Vero cells (should be onto Vero cells) and the cells were infected at an MOI of 0.001. The latter sentence could be revised to read, “Then, 500 mL of DV4 inoculum at a MOI of 0.001 was inoculated onto Vero cells”.
Section 2.6 The glucose levels in the culture medium were detected (determined could be a better word here ) using the GlucCellTM detector. Also, for section 2.7, “The pH levels in the culture medium were detected using a pH meter”. Again a better word for detected would be “determined” because the levels of glucose and the pH are being determined not just detected.
Section 2.8 What is meant by “Two pieces of the carriers and spell-out CVD and then abbreviate, and how was shock performed, in other words, shocked with what” in the following sentence,
“Two pieces of the carriers were taken out and placed into a 1.5mL centrifuge tube with 1mL of CVD cell lysate. The tube was shaken for 1 minute using a vibrator, and the carrier was then placed in an incubator at 37℃ for 1 hour with intermittent shocks every 15 minutes to release the nuclei. Next, 20μL of the suspension was added to the cell counting plate for cell counting (Countstar biotech)”.
The contents of this section 2.8 is not readily understood, as mentioned, “Two pieces of the carriers were taken out of what” and then placed in 1.5 ml centrifuge tubes” and then what, and then “1 ml of CVD cell lysate” which need clarification. Should this be a continuation of section 4?
Section 2.9 Suggest that the first sentence read, “The virus suspension (rather than fluid) and carrier were collected in 1.5 mL tubes daily. Also, the following sentence read, “The virus suspension (rather than fluid) was centrifuged at 10,000 g for 30 minutes to remove cell debris and obtain pure virus supernatant.
Section 2.11 Animal immunization, in this section, the contents conveys that the DV4 (should be DENV4) inactivated vaccine was obtained by heating the virus in a 56℃ water bath for 30 minutes and mixing it with TH-Z93 adjuvant suspended in PBS to achieve a final injection volume of 200μL. Female BALB/c mice, 6 weeks old, were housed in special cages and acclimatized to the environment for 7 days before immunization. The mice were immunized with the DV4 (should be DENV4) inactivated vaccine three (should be 3 times) times at two-week intervals. Each time, 100 μg of the corresponding DV4 (should be DEN4) inactivated vaccine was injected subcutaneously (should abbreviate as SC) through the neck. Blood samples (30 μL) were collected from the tail vein before each immunization and on day 14 after the third immunization. Control mice were injected with the same volume of PBS or adjuvant reagent. The serum was obtained by centrifuging the blood at 4000 rpm (convert to g) for 10 minutes in at 4℃. (delete “a centrifuge cooled at 4℃ (Eppendorf))”.
In this Section 2.11, nothing is mentioned about the vaccine regarding the concentration used along with the concentration of the adjuvant to inoculate the mice, what is the source and provide a brief description of the of the adjuvant? It is stated that, “The mice were immunized with the DV4 inactivated vaccine three times at two-week intervals”, how many time were the mice immunized?
Comments on the Results Section
According to the Material and Methods Section. “The protein concentration was measured using the Enhanced BCA Protein Assay Kit (Beyotime, P0010S) following the manufacturer's instructions”. However, it is not clear as to what was the concentration of the protein content of inactivated DENV4 concentration? Although the nucleic acid and the infectivity titer was determine in the infectious DENV4, the nucleic acid and protein content was not determined for the inactivated DENV4. As the authors know, the protein content of a potential vaccine is the determinant of the immune response.
The immune response of mice injected with the DENV4 was assessed using an ELISA for determining the IgG antibody. A more appropriate assessment of antibody should have been a neutralization assay as this is the more conclusive estimate of the protective efficacy of immune elicited protective neutralizing antibody.
Section 3.3 of the Results Section. The following sentence should be revised to read as follows: “Additionally, the antiserum from mice immunized with the DENV4 (delete DV) vaccine (cross reacted ) and delete (could cross-react) with both the inactivated virus from the bioreactor and the petri dish (Figure 6B-6G). These results indicate (indicated rather than indicate) that the bioreactor-derived DENV4 exhibited immunogenicity in BALB/c mice and had better immune effect than the conventionally petri dish-derived DENV4
Reviewer’s Overall Observations
Overall, the contents of this manuscript based on an adequately describe rationale that a more efficacious dengue vaccine is needed, however, this is misleading to the reader as the objective of the study is focused on the evaluation of a bioreactor for supporting the enhancement of the propagation of Vero cells for use to support the replication of DEN4 vaccine as a potential vaccine candidate. As, such the title needs to be changed from “Celcradle™ Bioreactor Enhances Dengue Vaccine Production and Immunogenicity” to reflect the objective and outcome of the study. Also, throughout the manuscript, dengue vaccine should be referred to as an intended vaccine and not as a vaccine as the evaluation of immunogenicity in mice does not warrant referring to the production of an inactivated DEN4 as being a vaccine in this study. The methods, results and discussion section as adequately described with the exception of the reviewer’s above comments. The methods employ appropriate technical procedures to yield valid data that in this study are scientifically sound. The discussion provides an acceptable interpretation of the main findings in relation to the results of other related studies. The conclusions are supported by the observation in this study that the author’s successfully employed a new tidal CelcradleTM bioreactor system to enhance the production of Vero cells that led to an increase in the yield of DEN4 virus for the potential development of candidate vaccines. The scientific merits of this study are suitable and scientifically sound enough to warrant further consideration for publication provided the authors consider the reviewer’s comments for improving the contents of the manuscripts.
The use of mice in this studies requires evidence that the protocols involving the use of mice was approved by an Institutional Animal Care and Use Committee or other ethical review board
Comments on the Quality of English LanguageThe quality of the use of the english language is acceptable, the main deficiency is the failure to use past tense as needed
Reviewer 2 Report
Comments and Suggestions for Authors
The authors evaluated vaccine antigen produced in a Celcradle bioreactor in mice. The authors successfully produced DV4 vaccine and demonstrated immunogenicity of them in mice. DV vaccine production is important area and this manuscript will also contribute the development of other viral vaccines. Comments for the authors below:
Major points:
1. Section 2.8: Please explain carriers and CVD.
2. Section 2.11: Please indicate the group size.
3. Section 2.12: Please indicate the coating volume and confirm the sample volume per well.
4. Figure 5: Please include the RNA results from Petri dish to compare the virus yields.
5. Figure 6: Please explain why antibody titers were dropped after the 2nd and 3rd immunizations.
6. Please determine the neutralizing antibody titers in immunized micel
Minor points:
1. “Celcradle™” should be “CelCradle™” throughout the manuscript.
2. Section 2.13: Please delete “and” after 9.5.
3. Figure 5: Please confirm the scale of Y-axis. Really Log10?
Round 2
Reviewer 1 Report
Comments and Suggestions for Authors
According to the plan of this study, the aim was to establish a process for the production of dengue whole virus vaccine in vero cells using the Celcradle™ bioreactor system and to assess the immunogenicity of the dengue virus antigen. While the results of the study demonstrated that the technology and procedure used were successful in producing high concentrations of dengue virus and that the virus elicited antibody in mice. thus demonstrating immunogenicity. However, as emphasized by this reviewer during his first review of the contents of the manuscript, there are no data to support that this technology and/or procedure could be used for the development and optimization in vaccine production. While the technology and procedure produced high concentration of dengue virus and the virus was immunogenic, these observations do not qualify as even a vaccine candidate. For example, the title of the manuscript, “Efficient Production and Immunogenicity of Dengue Virus Inactivated Vaccine Using the CelCradleTM Bioreactor System” is misleading as it does not represent the aim and results of the study which were to evaluate a bioreactor system for supporting a cell culture substrate for replicating high concentrations of dengue virus and to evaluate the immunogenicity. The authors continue to convey misleading statement, such as in section 3.2 entitled, “DENV proliferation through Vero cells in the CelCradleTM bioreactor system”, that, “After adapting Vero cells to suspension growth in the CelCradleTM bioreactor system, we proceeded to mass-produce the DENV vaccine” which is not correct, as the observations generated by this study does not support the development of a vaccine. On the other-hand, the last sentence of section 3.2 is the correct interpretation of the findings of this study, “These results demonstrated that the Vero cells adapted to suspension culture through the CelCradleTM bioreactor system can successfully produce DENV at a high titer, paving the way for the development of a bioreactor process for DENV vaccine production”. However, as another example of the conveyance of a mixed outcome of the study are the conclusions of the study, as follows, “In this study, we successfully established efficient DENV production using the new tidal CelcradleTM bioreactor system, a scalable cell culture system that can be used for process development and optimization in vaccine production. We demonstrated that the immunogenicity of the DENV vaccine derived from the CelcradleTM bioreactor system was excellent, and even superior to the petri dish-derived DENV vaccine", thus, misleading the reader to believe that the study developed vaccines.
The authors must avoid conveying the misleading information that this study developed a dengue vaccine and focus on the aim that was to establish a process for the production of dengue whole virus vaccine in vero cells using the Celcradle™ bioreactor system and to assess the immunogenicity of the dengue virus antigen.Therefore, nothing to do with a vaccine at this stage of the study, yes, this may come later as the increased dengue virus yield could be the answer to developing a successful dengue virus vaccine.
Comments on the Quality of English LanguagePast tense should be used as needed
Author Response
Response to Reviewer 1 Comments
Thank you very much for taking the time to review our manuscript. Please find the detailed responses below and the corresponding revisions highlighted in the re-submitted manuscript.
Comments on moderate editing of English language required and past tense should be used as needed
Response 1: Thank you for your valuable feedback. We have addressed the need for moderate editing of the English language in our manuscript. The revisions, including language edits, can be clearly seen in the submitted revision tracking mode manuscript. We have carefully reviewed each section and made appropriate changes to enhance the clarity and readability of the text and used past tense as neededin the manuscript.
We appreciate your attention to detail and are committed to ensuring the quality of our work. If you require any further information or have additional comments, please do not hesitate to let us know.
Comments on the conclusions supported by the results.
Response 2: Thank you for your insightful comments. We have carefully reconsidered the results section of our manuscript and have rewritten it to provide a more accurate and concise description of our findings. Additionally, we have revised the title to better reflect the outcomes of our study, ensuring that it aligns with our conclusions.
We believe these changes strengthen the manuscript and enhance the clarity of our results. We appreciate your attention to these matters and are grateful for your guidance in improving the quality of our work.
Please let us know if there are any further adjustments or additional information required.
Comments and Suggestions for Authors
According to the plan of this study, the aim was to establish a process for the production of dengue whole virus vaccine in vero cells using the Celcradle™ bioreactor system and to assess the immunogenicity of the dengue virus antigen. While the results of the study demonstrated that the technology and procedure used were successful in producing high concentrations of dengue virus and that the virus elicited antibody in mice. thus demonstrating immunogenicity. However, as emphasized by this reviewer during his first review of the contents of the manuscript, there are no data to support that this technology and/or procedure could be used for the development and optimization in vaccine production. While the technology and procedure produced high concentration of dengue virus and the virus was immunogenic, these observations do not qualify as even a vaccine candidate. For example, the title of the manuscript, “Efficient Production and Immunogenicity of Dengue Virus Inactivated Vaccine Using the CelCradleTM Bioreactor System” is misleading as it does not represent the aim and results of the study which were to evaluate a bioreactor system for supporting a cell culture substrate for replicating high concentrations of dengue virus and to evaluate the immunogenicity. The authors continue to convey misleading statement, such as in section 3.2 entitled, “DENV proliferation through Vero cells in the CelCradleTM bioreactor system”, that, “After adapting Vero cells to suspension growth in the CelCradleTM bioreactor system, we proceeded to mass-produce the DENV vaccine” which is not correct, as the observations generated by this study does not support the development of a vaccine. On the other-hand, the last sentence of section 3.2 is the correct interpretation of the findings of this study, “These results demonstrated that the Vero cells adapted to suspension culture through the CelCradleTM bioreactor system can successfully produce DENV at a high titer, paving the way for the development of a bioreactor process for DENV vaccine production”. However, as another example of the conveyance of a mixed outcome of the study are the conclusions of the study, as follows, “In this study, we successfully established efficient DENV production using the new tidal CelcradleTM bioreactor system, a scalable cell culture system that can be used for process development and optimization in vaccine production. We demonstrated that the immunogenicity of the DENV vaccine derived from the CelcradleTM bioreactor system was excellent, and even superior to the petri dish-derived DENV vaccine", thus, misleading the reader to believe that the study developed vaccines.
The authors must avoid conveying the misleading information that this study developed a dengue vaccine and focus on the aim that was to establish a process for the production of dengue whole virus vaccine in vero cells using the Celcradle™ bioreactor system and to assess the immunogenicity of the dengue virus antigen.Therefore, nothing to do with a vaccine at this stage of the study, yes, this may come later as the increased dengue virus yield could be the answer to developing a successful dengue virus vaccine.
Response 3: We sincerely appreciate your attention to the misleading descriptions in our manuscript. To address this concern, we have thoroughly reviewed and revised all relevant sections of the paper to ensure that the information presented is accurate and not prone to misinterpretation.
These modifications have been made to eliminate any potential for confusion or error in understanding our work. The revised manuscript has been submitted with tracked changes, allowing for a clear comparison between the original and updated text. We believe these revisions significantly improve the manuscript. We thank you for bringing this matter to our attention and for your valuable input.
Please check the attachment for specific changes.

Reviewer 2 Report
Comments and Suggestions for Authors
Response 1: Thank you for pointing this out. We have rivesed “CelCradle™” problem throughout the manuscript.
I still see a lot of ”Celcradle” in the manuscript.
Author Response
Response to Reviewer 2 Comments
Thank you very much for taking the time to review our manuscript. Please find the detailed responses below and the corresponding revisions highlighted in the re-submitted manuscript.
Comments 1: English language editing
Response 1: Thank you for your feedback regarding the need for English language editing in our manuscript. We have taken your comments to heart and have engaged in a thorough editing process to enhance the language and clarity of our paper.
The revised manuscript, submitted with tracked changes, reflects these editorial improvements. It is our hope that these edits have addressed the concerns raised and have elevated the overall quality of the writing.
We appreciate your attention to detail and your efforts to help us improve our work. If there are any additional comments or suggestions you have for further refinement, please do not hesitate to share them with us.
2. Questions for General Evaluation |
Reviewer’s Evaluation |
Response and Revisions |
Is the research design appropriate? |
Can be improved |
We have re-described the methodology, results, and discussion sections and revised the title to better align with the outcomes and conclusions of my study. |
Are the methods adequately described? |
Can be improved |
We have carefully addressed your comments by re-describing the methodology sections of my manuscript. |
Are the results clearly presented? |
Can be improved |
We have carefully addressed your comments by re-describing the result sections of my manuscript. |
Are the conclusions supported by the results? |
Can be improved |
We have carefully addressed your comments by re-describing the conclusion sections of my manuscript. |
Comments3: “Celcradle™” should be “CelCradle™” throughout the manuscript, and Still see a lot of ”Celcradle” in the manuscript.
Response 3: Thank you for pointing out the inconsistency in the naming convention for "Celcradle™." We have addressed this issue and ensured that the correct name "CelCradle™" is used consistently throughout the manuscript in our revised submission.
We appreciate your attention to detail and are committed to maintaining the accuracy of our work. If there are any further concerns or additional corrections needed, please feel free to inform us.
Please check the attachment for specific changes.
